# Efficacy and Safety of Gel Immersion Endoscopic Mucosal Resection for Non-Pedunculated Colorectal Polyps

**DOI:** 10.3390/life13030711

**Published:** 2023-03-06

**Authors:** Hiroshi Ashizawa, Kinichi Hotta, Kenichiro Imai, Sayo Ito, Yoshihiro Kishida, Kazunori Takada, Taishi Okumura, Noboru Kawata, Masao Yoshida, Yuki Maeda, Yoichi Yamamoto, Tatsunori Minamide, Junya Sato, Hirotoshi Ishiwatari, Hiroyuki Matsubayashi, Hiroyuki Ono

**Affiliations:** Division of Endoscopy, Shizuoka Cancer Center, 1007 Shimonagakubo, Nagaizumi, Shizuoka 411-8777, Japan

**Keywords:** colorectal cancer, gel immersion endoscopic mucosal resection, non-pedunculated colorectal neoplasms

## Abstract

Underwater endoscopic mucosal resection (UEMR) has become a popular endoscopic resection method for large colorectal neoplasms. However, visualization can be poor during UEMR due to the presence of intestinal fluid. Gel immersion endoscopic mucosal resection (GIEMR), using a specially developed gel (Viscoclear^®^, Otsuka Pharmaceutical Factory, Tokushima, Japan), can improve the visual field. However, reports of GIEMR for colorectal polyps are limited. Herein, we evaluated the short-term outcomes of GIEMR for non-pedunculated colorectal neoplasms (NPCRN). This single-center, retrospective, and observational study includes 25 lesions in 20 patients with NPCRN who underwent GIEMR between January and October 2022. The short-term outcomes and adverse events were evaluated. The lesion locations were as follows: right colon, 18 lesions; left colon, 7 lesions; and rectum, none. The median tumor diameter was 15 (IQR, 10–18) mm. Histological classification was as follows: sessile serrated lesion, 9 cases; adenoma, 12 cases; and intramucosal adenocarcinoma, 4 cases. The overall en bloc resection rates and R0 resection rates were 80% (20/25) and 72% (18/25). For NPCRN in 10–19 mm, the en bloc resection rate was 75% (12/16), with an R0 resection rate of 69% (11/16). No post-polypectomy bleeding, perforation, or post-coagulation syndrome were observed. The findings of our study provide preliminary evidence of the efficacy and safety of GIEMR for NPCRN. Therefore, GIEMR may be a promising novel endoscopic resection method for NPCRN.

## 1. Introduction

As colorectal adenomas are considered precursor lesions for colorectal cancer, the endoscopic resection of them can reduce future colorectal cancer mortality [1].

Recently, serrated lesions of the colon have been proposed as a carcinogenic pathway, via the serrated neoplastic pathway, accounting for 15–30% of all colorectal cancers [2]. Based on these findings, colorectal adenomas and serrated lesions are indications for endoscopic resection.

For the endoscopic resection of non-pedunculated colorectal neoplasms (NPCRNs), cold snare polypectomy (CSP), as a non-electrosurgical resection method, has been reported to have a non-inferior rate of complete histological resection [3] and less post-polypectomy bleeding [4] compared to hot snare polypectomy (HSP) for lesions <10 mm. Therefore, the American Society of Gastrointestinal Endoscopy (ASGE) and the European Society of Gastrointestinal Endoscopy (ESGE) guidelines recommend CSP for lesions <10 mm [5,6]. On the other hand, in addition to CSP, HSP and endoscopic mucosal resection (EMR) are also recommended for 10–19 mm NPCRNs according to the ASGE. Meanwhile, the ESGE guidelines recommend HSP and EMR for 10–19 mm NPCRNs.

Incomplete resection is a risk factor for residual recurrence in endoscopic resection of colorectal lesions [7,8]. Therefore, complete resection is important for avoiding recurrence. However, according to previous reports, the current resection methods are not adequate, especially in larger or serrated lesions. A retrospective study of EMR in 572 colorectal lesions >10 mm reported 23.5% of residual recurrence in cases of incomplete resection [8]. A retrospective study of EMR revealed a 45% incomplete resection rate in 101 sessile serrated lesions (SSL) >10 mm [9]. Taken together, these findings show that conventional endoscopic mucosal resection (CEMR), which uses submucosal injection, is associated with a high rate of incomplete resection for SSLs and adenomas >10 mm [10]. Therefore, it has been desirable to develop endoscopic resection methods that can reduce incomplete resections.

Recently, underwater endoscopic mucosal resection (UEMR), which does not require submucosal injection, has become popular in endoscopic practice [11,12,13,14]. A multicenter, randomized controlled trial comparing UEMR to CEMR for NPCRNs 10–20 mm in size reported significantly superior outcomes for UEMR, with an en bloc resection rate of 89% for UEMR compared to 75% for CEMR, with an R0 resection rate of 69% and 50%, respectively [15]. On the other hand, during water immersion, the visual field may become poor due to dirty intestinal fluid resulting from poor bowel preparation. By contrast, gel immersion endoscopic mucosal resection (GIEMR), uses an endoscopic field-enhancing gel (Viscoclear^®^, Otsuka Pharmaceutical Factory Co., Tokushima, Japan) to maintain a clear visual field despite intestinal fluid or bleeding [16,17,18]. The usefulness of GIEMR has been reported for endoscopic resection in the stomach, duodenum, and colon [19,20,21]. However, evidence on the use of GIEMR for colorectal polyps is limited to a few case reports [22]. As discussed above, piecemeal or incomplete resection with CEMR is problematic for NPCRNs. On the other hand, in most cases, pedunculated-type polyps can be resected en bloc with HSP, and the existing treatments are less problematic. In other words, GIEMR is rarely performed for pedunculated-type polyps, and polyp resectability and safety for NPCRNs are the most important points to consider. Therefore, we excluded pedunculated-type polyps in this study. Our aim in this study was to evaluate the short-term outcomes and adverse events of GIEMR for NPCRNs.

## 2. Materials and Methods

### 2.1. Study Population and Statement of Ethics

This was a single-center, retrospective, and observational study including 20 patients with NPCRNs, contributing 25 lesions, who underwent GIEMR at our hospital between January and October 2022. NPCRNs ≥ 10 mm with a diagnosis of non-invasive adenocarcinoma based on gross appearance and narrow-band imaging (NBI) findings by magnified color endoscopy were included. However, suspected intramucosal adenocarcinomas were considered for resection even if they were <10 mm. The gross appearance was made according to the Paris endoscopic classification [23]. The Paris endoscope classification has six types including the following: Type 0: superficial polypoid, flat/depressed, or excavated tumors; Type 1; polypoid carcinomas, usually attached on a wide base; Type 2: ulcerated carcinomas with sharply demarcated and raised margins; Type 3: ulcerated, infiltrating carcinomas without definite limits; Type 4: non-ulcerated, diffusely infiltrating carcinomas; and Type 5: unclassifiable advanced carcinomas. Type 0 lesions are classified into three distinct groups: type 0-I, polypoid; type 0-II, non-polypoid and non-excavated; and type 0-III, non-polypoid with a frank ulcer. The subgroups type 0-I and 0-II are then segmented again. Type 0-I includes two variants with pedunculated (0-Ip) and sessile (0-Is). Type 0-II includes three variants that are slightly elevated (0-IIa), completely flat (0-Iib), and slightly depressed without ulcer (0-IIc). Type 1–5 were suggestive of invasive cancer and were excluded in this study. In addition, pedunculated-type polyps (Type0-Ip) can be resected en bloc with HSP, and existing treatments are less problematic. Therefore, we excluded pedunculated-type polyps in this study. NBI diagnosis was made according to the Japan NBI Expert Team (JNET) classification [24]. That is, normal/hyperplastic lesions/sessile serrated polyps were classified as JNET Type 1, low-grade intramucosal neoplasia including intramucosal cancer with low-grade structural atypia as JNET Type 2A, high-grade intramucosal neoplasia/shallow submucosal invasive cancer as JNET Type 2B, and deep submucosal invasive cancers as JNET Type 3. Findings with invisible micro-vessel patterns and regular dark or white spots, similar to surrounding normal mucosa, were classified as JNET Type 1. Findings such as (i) regular caliber vessel pattern or regular distribution variable as meshed/spiral patterns and (ii) regular surface patterns as tubular/branches/papillary were classified as JNET Type 2A. Findings such as (i) variable caliber of vessels; (ii) irregular distribution of vessels; and (iv) irregular or obscure surface patterns were classified as JNET Type 2B. Findings such as (i) loose vessel areas; (ii) interruption of thick vessels; and (iii) amorphous areas of surface patterns were classified as JNET Type 3. Lesions with endoscopic findings suggestive of JNET Type 3 were excluded from resection. The resection method was decided to be CEMR, UEMR, or GIEMR at the endoscopist’s discretion. Each resection method is described as follows. CEMR: Distal hood was attached to the endoscope, after needle injection of normal saline into the submucosa, and the lesion was strangulated by snare wire closure and resected using an electrosurgical unit. UEMR: after complete deflation of the colorectal lumen, total immersion of the lesion in normal saline using a mechanical water pump, snaring the lesion and the surrounding mucosa, and resection using electrocautery. GIEMR was described in detail in Section 2.3. GIEMR was performed by five endoscopists, four expert endoscopists, and one non-expert endoscopist. Expert endoscopists were certified by the Japan Gastroenterological Endoscopy Society. This study was approved by the Institutional Review Board of Shizuoka Cancer Center (J-2022-83).

### 2.2. Outcome Measures

The primary outcome was the R0 resection rate. The secondary outcomes were the en bloc resection rate, procedure time, and adverse events. R0 resection was defined as en bloc resection with a free pathological surgical margin. En bloc resection was defined as the endoscopic single-piece removal of the lesion. Procedure time was defined as the time from the start of gel immersion to complete lesion resection. Adverse events were graded according to the Common Toxicity Criteria for Adverse Events v5.0. Perforation during the procedure was defined as the presence of peritoneal fat on endoscopic imaging or leakage of air/luminal contents outside the gastrointestinal tract on abdominal computed tomography. Post-polypectomy bleeding was defined as bleeding requiring endoscopic hemostasis within 30 days of GIEMR. Post-coagulation syndrome was defined as abdominal pain, accompanied with a high-grade fever, and without perforation. A subset analysis was performed regarding lesion location (right-sided, left-sided, or rectal), morphology, lesion size, and the Japan narrow-band imaging expert team (JNET) classification.

### 2.3. Endoscopic Procedures and Equipment

All patients underwent same-day bowel preparation prior to colonoscopy, using a magnesium citrate solution or polyethylene glycol solution. Before the start of the colonoscopy, intravenous butyl bromide or glucagon and pethidine were administered to suppress intestinal peristalsis and reduce pain.

A colonoscope with a magnification function (PCF-H290ZI or CF-EZ1500DI; Olympus Co., Tokyo, Japan) was used in all cases.

A distal attachment (F-030; Top Co., Tokyo, Japan; MAJ-2187, MAJ-2257; Olympus Co., Tokyo, Japan) was attached to the tip of the scope and a non-traumatic catheter (Olympus Co., Tokyo, Japan) was used for observation and measurement of the lesions. The morphology of lesions was defined according to the Paris classification. Endoscopic characterization, using NBI with magnification according to the JNET classification, was performed in all cases. If the border of the lesion was difficult to determine, 2.1% of acetic acid was sprayed on the lesion, which was more often required for serrated lesions. Before GIEMR, a BioShield Irrigator (Fujifilm Medical, Tokyo, Japan) was attached to the forceps channel of the scope. The gel (Viscoclear^®^) aspirated in a syringe was injected through the BioShield Irrigator. The amount of gel (Viscoclear^®^, Otsuka Pharmaceutical Factory, Tokushima, Japan) was 200 g per bag, and the price per bag was JPY 2000 (USD 15.4).

Before gel immersion, retention gas in the colonic lumen was kept as low as possible. Gel injection was performed until a good visual field of the target lesion was obtained. A polypectomy snare, 15 mm or 20 mm in size (CaptivatorII^®^; Boston Scientific Japan Co., Tokyo, Japan/SnareMasterPlus^®^; Olympus Co., Tokyo, Japan), was used for resection owing to its good expandability. The VIO 300D electrosurgical unit was used for resection (Erbe Elektomedizn GmbH, Tübingen, Germany), with the following blend setting: endocut Q; effect 3; duration 2; and interval 2 (Table 1). During the procedure, additional gel was injected as necessary to maintain a clear visual field. The treatment procedure using GIEMR is shown (Figure 1 and Appendix A). When a residual lesion was recognized, additional snaring was used to achieve complete resection. Resected lesions were retrieved using forceps or nets. After endoscopic resection was performed, the wound was evaluated using NBI to rule out any residual tumor. Clip closure of artificial ulcers after resection was performed as necessary, based on endoscopists’ preference. The single tissue specimen obtained after en bloc resection was gently stretched. The specimen was pinned on a soft, porous material with the mucosal surface up and placed in 10% formalin. For the lesions removed with a piecemeal technique, the endoscopist reconstructed the entire lesion surface on soft, porous material from the fragments. The surface of the fixed specimen was examined and photographed. In the pathology laboratory, the specimen was withdrawn from the fixative and pre-cut in parallel fragments, 2 mm in width for GIEMR specimens. The margins in the adjacent normal mucosa were included for analysis in the serial histologic sections. The pathologist evaluated the histology and assessed the degree of differentiation of the tumor, the depth of invasion, and the completeness of excision. The resection was complete if the margins of the specimen were free from tumor tissue on serial sections. This concerns all the margins (vertical and lateral) on GIEMR specimens. Adverse events (post-polypectomy bleeding, perforation, and post-coagulation syndrome) were confirmed at outpatient visits, which were performed approximately one month post-GIEMR.

## 3. Results

### 3.1. Patients and Polyp Characteristics

The baseline characteristics of the 20 patients and their 25 NPCRNs included in the study are reported in Table 2. The median age of the study group was 70 (range, 53–90) years, with 14 patients being male (70%) and 2 patients (10%) taking antithrombotic medications. The median lesion size was 15 (IQR, 10–18; range, 6–26) mm and the dominant location was the right colon (72%). Twenty lesions (88%) were of the superficial elevated type (0–IIa). Nine lesions were diagnosed as JNET Type 1 and sixteen as Type 2.

### 3.2. Outcomes

The short-term outcomes are reported in Table 3 and Table 4. The overall R0 resection rate was 72% and the overall en bloc resection rate was 80%. The R0 resection rates according to size were 100% for 6–9 mm, 69% for 10–19 mm, and 60% for 20–26 mm. The R0 resection rates according to histological type were 75% for adenomas, 75% for intramucosal adenocarcinomas, and 67% for SSL. The en bloc resection rates according to size were 100% for 6–9 mm, 75% for 10–19 mm, and 80% for 20–26 mm. The en bloc resection rates according to histological type were 83% for adenomas, 100% for intramucosal adenocarcinomas, and 67% for SSL. After the endoscopic resection, the wound was evaluated using NBI and no residual tumor was found in all cases, so no case underwent adjuvant thermal ablation to the margin. All resected specimens included submucosal tissue based on the pathological diagnosis. The median resection time was 195 (range, 33–977) s. Conclusive pathological diagnoses of resected lesions included 9 serrated lesions, 12 adenomas, and 4 intramucosal adenocarcinomas. As for the amount of gel used, in nineteen cases (76%), one bag was enough to fill the colon lumen and perform GIEMR. In six cases (24%), when peristalsis of the intestine made it difficult to maintain a clear visual field, two bags of gel were used and frequent additional injections were necessary.

### 3.3. Adverse Events

There was no incidence of post-polypectomy bleeding, intraprocedural perforation, or post-coagulation syndrome in the 1-month follow-up after GIEMR.

## 4. Discussion

To the best of our knowledge, this is the first study to investigate the efficacy and safety of GIEMR for NPCRNs. There have only been a few case reports on the use of GIEMR for colorectal lesions and, thus, its efficacy and safety have not been verified. In this study, the overall en bloc resection rate using GIEMR was 80%, with an R0 resection rate of 72%, with no adverse events. For NPCRNs in 10–19 mm, the en bloc resection rate was 75%, with an R0 resection rate of 69%.

The current standard resection method is CEMR, with an R0 resection rate of 50–77% for colorectal lesions 10–19 mm in size [8,10]. A multicenter, prospective randomized controlled trial comparing UEMR and CEMR for colorectal lesions 10–19 mm in size reported a significantly better R0 resection rate for UEMR (69%) than CEMR (50%) [15]. The R0 resection rate of GIEMR in our study was comparable to that of UEMR. In a previous study [9], the R0 resection rate for SSLs >10 mm was 45% for CEMR, compared to our rate of 67% for GIEMR. Although a direct comparison of our findings to previous studies is difficult due to the differences in backgrounds, the results of our study do support GIEMR as a promising novel endoscopic resection method for NPCRN.

In endoscopic resection, incomplete resection is a risk factor for local recurrence, with complete resection being important to prevent recurrence. Klein et al. [25] showed that the ablation of mucosal margins after endoscopic en bloc resection significantly reduced local recurrence, suggesting that tumor cells may remain at the resected margins. This means that even if en bloc resection was obtained, there were a certain number of cases with incomplete resection. Therefore, we considered the R0 resection rate to be a better surrogate marker for recurrence than the en bloc resection rate, and this was the primary outcome in our study. A characteristic result of this study is that there is little difference between the en bloc resection rate and the R0 resection rate. Usually, the R0 resection rate tends to be lower than the en bloc resection rate. One reason for this is that the burn effect on the resected specimen edge makes it difficult to assess the horizontal margins. In GIEMR, we postulate that the heat sink effect of gel immersion generates a more targeted application of the current while protecting against the burn effect to the resected specimens [11].

Paul et al. [8] reported that SSLs were more likely to be incompletely resected than other neoplastic polyps, with almost half (47.6%) of all large (10–20 mm) SSLs being incompletely removed, likely due to the low visibility of the lesion margin. Therefore, we evaluated the R0 resection rate separately for adenomas and SSLs. In our study, The R0 resection rates of SSL were 67%, compared to 75% for adenomas, which was not as large a difference as previously reported. The reason for this was thought to be that there was no increase in the lesion size because there was no submucosal injection, and the buoyancy effect of the gel, which facilitated snaring. Although the R0 resection rate of SSL was higher than previously reported [8], a large number of cases would need to be evaluated.

The adverse events of CEMR that were previously reported include a rate of 1.1–1.7% for post-polypectomy bleeding and 0.5–0.8% for perforation [26,27,28,29]. In our study, no adverse events were identified post-GIEMR. UEMR is a new endoscopic resection method that has attracted considerable attention in recent years, with air deflation and water immersion used to lift and float the lesion making it easier to capture the lesion. As submucosal injection is not required with these techniques, the wound after endoscopic resection is smaller and clip suturing is easier. Similarly, GIEMR has advantages over CEMR, including the easier capture of lesions and a smaller post-resection wound. In addition, as gel is more viscous than water, it is less likely to become turbid, allowing a clear visual field to be maintained for a longer period of time, and if the gel does become turbid, a clear visual field can easily be regained by the insertion of additional gel, which is a useful advantage of GIEMR. Moreover, the BioShield Irrigator allows for additional gel insertion without removing the snare.

The gel immersion method was firstly reported by Yano T et al. who indicated that gel immersion provides a clear visual field for poor vision caused by blood or intestinal fluid underwater [16]. Yano T et al. indicated that gel immersion is effective in patients when it is difficult to secure the visual field using air insufflation or water immersion because gel immersion creates a space for endoscopic visualization and treatment. Additionally, gel immersion is effective in patients with ongoing bleeding in a narrow lumen such as the esophagus or small or large bowel. Since then, the benefits of GIEMR have been reported. Although not colorectal, a retrospective study comparing GIEMR and UEMR for the resection of a superficial non-ampullary duodenal epithelial tumor reported a superior R0 resection rate for GIEMR, with a shorter procedure time [30,31]. According to these studies, GIEMR can reduce the procedure time compared to UEMR because gel facilitates a clear visual field more rapidly. In UEMR, when it is difficult to immerse the lesion in water due to air bubbles or water outflow, injecting a large amount of water must be performed several times. On the other hand, gel is viscous, which easily removes air bubbles and remains in the lumen. It is suggested that this is the reason for the shorter execution time. In our study, we examined the short-term results of GIEMR for NPCRNs, but they did not directly compare to CEMR or UEMR. However, compared to previous reports, the R0 resection rate for 10–19 mm NPCRNs was comparable to UEMR. The results indicate that GIEMR has a high safety and R0 resection rate, and can be considered as a promising novel endoscopic resection method for NPCRNs.

The use of gel must also be considered in cost economics. Each gel bag contains 200 g, and the price per bag is JPY 2000 (USD 15.4). Since this is a retrospective study, it is difficult to assess the exact amount of gel used; for instance, nineteen cases (76%) were able to complete GIEMR with only one bag, but in six cases (24%), frequent gel injection was required and two bags of gel were used. The main reason for the increased gel use was the difficulty in maintaining a clear visual field due to strong intestinal peristalsis. In a previous study on GIEMR for superficial non-ampullary duodenal epithelial tumors, the median amount of gel required to immerse the lesion was only 100 mL, which was significantly less than the amount of water required (330 mL) in another study on UEMR [29]. The amount of gel used tends to be higher in the colon than in the duodenum because the colon has a larger lumen and more intestinal fluid than the duodenum. Because of the extra cost of one–two bags of gel compared to UEMR, GIEMR should be performed in selected cases where it is more effective than UEMR. However, previous studies have yet to suggest which cases should be treated with GIEMR, which is an issue for the future. GIEMR is a new endoscopic resection technique, and there is no report yet on its use on large colorectal polyps. Therefore, we consider it necessary to first examine the efficiency of GIEMR and publish these preliminary results. If the superior results of GIEMR compared to UEMR can be proven in many cases, the cost of the gel will be worth it. The limitations of our study need to be acknowledged. First, this was a retrospective study conducted at a single center and, therefore, the potential for bias in the selection of patients and lesion and the proficiency level of endoscopists cannot be denied. To solve this problem, a study involving many physicians and institutions is desirable. Second, the exact amount of gel injected was not recorded in many cases and, therefore, an accurate cost assessment could not be performed. Third, the study sample was limited. Larger samples would be desirable in future studies for an accurate evaluation of the adverse event rate. Lastly, the follow-up period was less than one year after GIEMR and, therefore, it was not possible to evaluate the local recurrence. A prospective observational study with a sufficient follow-up period is required to provide high-quality evidence for GIEMR.

## 5. Conclusions

Findings of an R0 resection rate of 72% for GIEMR in our study sample of 25 NPCRN lesions with no adverse events, provides preliminary support for GIEMR. Therefore, GIEMR may be a promising novel endoscopic resection method for NPCRNs.

## Figures and Tables

**Figure 1 life-13-00711-f001:**
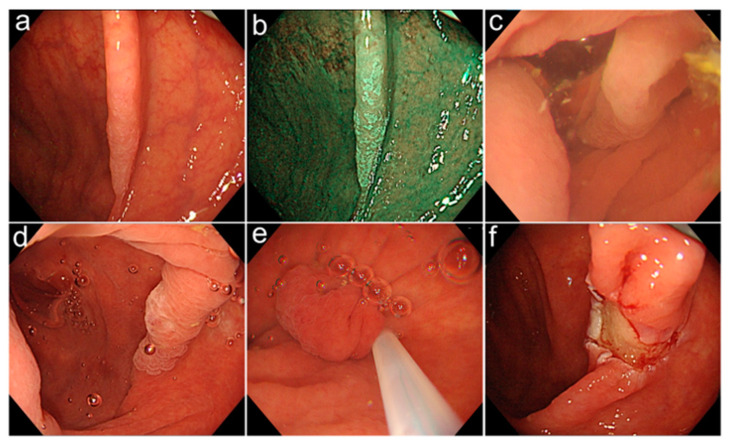
(**a**) White light endoscopy showing a 13 mm flat elevated lesion in the ascending colon. (**b**) Narrow-band imaging (NBI) shows that the lesion is white in color, with no micro-vessels observed. The lesion was diagnosed as a sessile serrated lesion (SSL). (**c**) Immediately after gel immersion, the lesion was first slightly turbid. (**d**) Following further immersion gel, a clear visual field was quickly obtained. (**e**) With a clear visual field, snaring was possible, with confirmation of the tumor margin. (**f**) The lesion was resected en bloc, without residual; the wound after resection is shown.

**Table 1 life-13-00711-t001:** Equipment and settings used for GIEMR in this study.

Scope	PCF-H290ZI/CF-EZ1500DI (Olympus Co., Tokyo, Japan)
Gel	Viscoclear^®^ (Otsuka Pharmaceutical Factory, Tokushima, Japan)
Attachment	Elastic touch^®^ (Top, Tokyo, Japan) Distal hood^®^ (Olympus, Tokyo, Japan)
Channel device	BioShield Irrigator^®^ (Fujifilm Medical, Tokyo, Japan)
Electrosurgical unit	VIO 300D ENDOCUT Q EFFECT 3/DURATION 2/INTERVAL 2
Snare	CAPTIVATORII^®^ (15/20 mm) (Boston scientific, Tokyo, Japan) SnareMater Plus^®^ (15 mm) (Olympus, Tokyo, Japan)

**Table 2 life-13-00711-t002:** Baseline characteristics of the patients and polyps.

No. of patients, n	n = 20
Gender, male	14
Median age, years (range)	70 (53–90)
Antithrombotic drugs	2
No. of lesions, n	n = 25
Location, n (%)	
Cecum	6 (24)
Ascending colon	8 (32)
Transverse colon	4 (16)
Descending colon	3 (12)
Sigmoid colon	4 (16)
Rectum	0 (0)
Morphology, n (%)	
0-Is/Isp	3 (12)
0-IIa	22 (88)
NBI magnifying findings, n (%)	
JNET Type 1	9 (36)
JNET Type 2A	14 (56)
JNET Type 2B	2 (8)
Median tumor size, mm (IQR)	15 (10–18)

NBI, narrow-band imaging. JNET, Japan narrow-band imaging expert team classification.

**Table 3 life-13-00711-t003:** Short-term outcomes of GIEMR.

	n = 25
En bloc resection, n (%)	20 (80)
R0 resection, n (%)	18 (72)
R1 resection, n (%)	5 (20)
RX resection, n (%)	2 (8)
Including submucosal tissue, %	100
Median resection time, sec (IQR)	195 (156–290)
Histological diagnosis, n (%)	
Adenoma	12 (48)
Tis	4 (16)
SSL	9 (36)
Complication, n (%)	
Perforation	0
Delayed bleeding	0
Post polypectomy coagulation syndrome	0

**Table 4 life-13-00711-t004:** Resection results according to tumor size and histological type.

	En Bloc Resection	R0 Resection
Tumor size, mm		
6–9	4/4 (100)	4/4 (100)
10–19	12/16 (75)	11/16 (69)
20–26	4/5 (80)	3/5 (60)
Histological type		
Adenoma	10/12 (83)	9/12 (75)
Tis	4/4 (100)	3/4 (75)
SSL	6/9 (67)	6/9 (67)

## Data Availability

The data presented in this study are available on request from the corresponding author. The data are not publicly available due to local data regulations.

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
