# Peer review of "Efficacy and Safety of Gel Immersion Endoscopic Mucosal Resection for Non-Pedunculated Colorectal Polyps"

_life, 2023, doi:10.3390/life13030711_

Round 1

Reviewer 1 Report

This is a unique method using biscoclear. I think this method could be applied in many ways. I hope that my comment is very useful for the improvement of the article.

1)      Why was the pedunculated type excluded from this study?

2)      Since this is a new method, the amount of gel used for GIEMR should be stated, even if it is a fractional example. It will probably be used more than the duodenum and would be helpful to many people.

Reviewer 2 Report

The authors conducted a single-center, retrospective, observational study on patients with NPCRN who underwent GIEMR to evaluate short-term outcomes and adverse events. The methodology was well-organized and appropriate, and the results were interesting and may provide valuable information for readers of this journal. However, several points should be further assessed as follows:

1.     Despite the authors' mention of the inability to perform an accurate cost assessment in the limitations section, the cost-effectiveness of this new resection method should still be discussed. If the GIEMR is more expensive than conventional UEMR, the authors need to clarify which patient should selectively undergo it. Do the authors think we should perform GIEMR for all cases or patients should be selected?

2.     Related to comment 1, it is necessary to mention the actual cost of the gel in terms of USD in the method section. Additionally, the amount of gel usually used per patient (even if the exact amount is not known) should also be noted.

3.     How many cases with lesions greater than 20mm had adjuvant thermal ablation of the post-EMR margin should be noted as ASGE guidelines suggest it.

4.     P2, L56. This description is incorrect. ASGE guidelines recommend not only HSP or EMR but also CSP to remove 10–19 mm non-pedunculated lesions. Please correct this description.

5.     Please add the ASGE and ESGE guidelines to the reference list and indicate the reference number in the appropriate part of the text.

Reviewer 3 Report

Thanks to the editors for the opportunity to review this paper. The work is very well written, brings interesting results, but I have a few minor comments.

The authors did not provide the proper references in 2nd paragraph of introduction. In addition, the other cited papers are well selected and valid.

A relevant limitation is the lack of long-term follow up, so it's hard to compare the methods. However, recurrence rate is an important factor when choosing treatment. Nevertheless, this is a good field for updating results in the future and perhaps the inclusion of more people and even centers.

Round 2

Reviewer 1 Report

This manuscript has been well revised.